# Effects of Denture Cleaning Regimens on the Quantity of *Candida* on Dentures: A Cross-Sectional Survey on Nursing Home Residents

**DOI:** 10.3390/ijerph192315805

**Published:** 2022-11-28

**Authors:** Yasuhiro Nishi, Katsura Seto, Mamoru Murakami, Kae Harada, Masakazu Ishii, Yuji Kamashita, Shinichiro Kawamoto, Tohru Hamano, Takuya Yoshimura, Asutsugu Kurono, Yasunori Nakamura, Masahiro Nishimura

**Affiliations:** 1Department of Oral and Maxillofacial Prosthodontics, Field of Oral and Maxillofacial Rehabilitation, Kagoshima University Graduate School of Medical and Dental Sciences, Kagoshima 890-8544, Japan; 2Seto Dental Clinic, Kagoshima 899-4101, Japan; 3Department of Removable Prosthodontics and Implant Dentistry, Advanced Dentistry Center, Kagoshima University Hospital, Kagoshima 890-8544, Japan; 4Department of Oral and Maxillofacial Surgery, Kagoshima University Graduate School of Medical and Dental Sciences, Kagoshima 890-8544, Japan; 5Saiseikai Kagoshima Hospital, Kagoshima 892-0834, Japan; 6Department of Dentistry and Oral Surgery, National Hospital Organization Kagoshima Medical Center, Kagoshima 892-0853, Japan

**Keywords:** oral care, denture cleaning regimen, nursing home residents, denture cleanser

## Abstract

Oral care involving a denture cleaning regimen is important for reducing the incidence of systemic diseases. However, limited information is currently available on denture cleaning frequencies and regimens. Therefore, the present study investigated the relationship between the number of *Candida* spp. present on the complete dentures of nursing home residents and cleaning regimens. Residents were surveyed to assess their denture cleaning methods. Plaque was collected by applying a sterile swab to the mucosal surface of each examined complete denture worn by 77 residents, and the *Candida* spp. collected were cultured, identified, and quantified. The relationship between denture cleaning regimens and the quantity of *Candida* spp. was investigated. Correlation and multivariable analyses revealed that the strongest factor influencing the number of *Candida* spp. on dentures was the frequency of use of denture cleansers. The number of *Candida* spp. was the lowest on dentures cleaned daily with a denture cleanser. The present results demonstrated that the daily use of a denture cleanser effectively controlled the adherence of *Candida* spp. to dentures. Oral and other healthcare providers need to provide instructions on and assist nursing home residents with the daily care of dentures, using denture cleansers, including the environment where cleaning is performed.

## 1. Introduction

The age-standardized prevalence and incidence of severe tooth loss and edentulism have declined at the global, regional, and country levels across a two-decade observation period [1,2]. However, since the average life of the elderly is being extended and the mean number of teeth lost increases with age, many individuals older than 75 years are still edentulous [3]. The prevalence of edentulism is affected by socio-educational and -economic levels, and it is higher among individuals with low education and income levels [3,4]. Therefore, many older individuals need complete dentures. Furthermore, the number of denture wearers increases when partial denture wearers are included.

Dentures placed in the mouth are colonized by a complex microbial plaque biofilm, which consists of numerous bacteria and fungi [5]. High numbers of *Candida* spp. have been detected on denture surfaces and the oral mucosa in dependent older adults [6,7,8]. Although *Candida* spp. are the microorganisms most commonly associated with denture stomatitis, colonization by *Candida albicans* is also closely linked to the development of serious diseases, such as aspiration pneumonia and mycosis of the esophageal mucosa [9], in immunocompromised individuals. Denture cleaning is an important aspect of systemic infective disease prevention, in addition to oral hygiene care, because effective oral care has been reported to reduce the incidence of systemic diseases, such as aspiration pneumonia and influenza, and promote the maintenance of and improvements in the quality of life of dependent older adults [10,11,12,13]. Dentures may function as a reservoir of oral microorganisms for systemic diseases [14,15,16,17]. Therefore, it is important to protect older adult denture wearers against microorganisms such as *Candida* spp. by including a denture cleaning regimen as an important aspect of oral hygiene care.

Denture hygiene may be achieved by using mechanical or chemical cleaning methods. Although many studies have been conducted on denture cleaning, limited information is currently available on specific cleaning-method regimens. Regarding chemical cleaning methods and/or regimens, evidence-based guidelines for complete dentures published in 2011 advocate daily denture soaking and brushing with an effective, non-abrasive denture cleanser; however, these guidelines provide no evidence for the beneficial effects of daily soaking [18]. A systematic review that examined the chemical cleaning methods used to reduce the numbers of *Candida* spp. on dentures was recently published and proposed effective chemical agents [19], but not methods or regimens using chemical cleansing. Under these conditions, reports have been published on regimens such as the frequency of use of denture cleansing agents. In an in vitro study, the use of a commercially available peroxide denture cleanser every 24 h was shown to inhibit mature biofilm formation by *C. albicans* [20]. In vivo studies suggested that the number of microorganisms adhering to complete dentures was influenced by denture cleaning methods, such as the use of a denture brush and the frequency of use of denture cleansers [8]. An in vivo and in vitro study recently demonstrated that daily denture cleansing regimens were superior to intermittent denture cleansing for reducing microbial numbers and plaque scores [21]. Moreover, a cross-sectional study on community-dwelling older adults indicated that the daily cleaning of dentures reduced the risk of pneumonia based on findings showing that infrequent denture cleaning was associated with the incidence of pneumonia [22].

However, in vitro and in vivo studies also reported that although chemical denture cleaning using peroxide-based effervescent cleansers decreased *Streptococcus* spp. and total bacterial levels, it was ineffective against *Candida* spp., including *C. albicans* [23,24,25,26,27,28]. Moreover, two in vivo studies showed that the daily use of peroxide-based effervescent cleansers was ineffective against *Candida* spp. but decreased *Streptococcus* spp. and total bacterial levels [29,30]. It is important to note that the immersion time in denture cleansers was only 3 min in these two studies.

Since the prevention of colony formation by microorganisms on dentures and the removal of any adhering microorganisms are important, further studies are needed to identify the types of cleaning procedures that prevent microbial adhesion to dentures. We planned to investigate the effects of the denture cleaning regimens used by nursing home residents on the number of *Candida* spp. in denture plaques in vivo because, even though the ineffectiveness of denture cleansers against *Candida* spp. has already been demonstrated [23,24,25,26,27,28,29,30], the in vivo relationship between the number of *Candida* spp. on dentures and denture cleaning regimens remains unclear. Therefore, it was hypothesized that the number of *Candida* spp. adhering to dentures may be related to the frequency of use of denture cleansers.

The purpose of the present cross-sectional study was to investigate the relationship between denture cleaning methods and/or regimens and the number of *Candida* spp. adhering to the dentures of nursing home residents.

## 2. Materials and Methods

### 2.1. Participants

A total of 77 edentulous individuals (21 males, 56 females; mean age, 84.4 years; S.D., 7.7 years; range, 68–102 years) agreed to participate in the present study. All participants were residents of a geriatric health service facility (nursing home residents) who required physical rehabilitation and care before being permitted to return home. They were treated in a general medical ward for residents with almost no decline in cognition. The aim of the present study was explained to the participants by using a document approved by the Clinical Study Ethics Committee of Kagoshima University Hospital (#18–89), and written consent was obtained from all participants. Participants used complete upper and/or lower dentures made of heat-cured acrylic resin without a metal base. They were satisfied with their dentures, cleaned them by themselves (based on interviews with the participants and their caregivers), and were free of oral disease, such as denture stomatitis and oral dryness (according to clinical examinations and medical records).

The dentures examined included 75 upper and 77 lower complete dentures (152 dentures in total) that had been worn by the nursing home residents and had not been relined; the surface roughness of some relined resins was found to increase over time when immersed in water [31]. In the G-Power software (Heinrich-Heine-Universität, Düsseldorf, Germany), the sample size of the multiple regression analysis with seven explanatory variables on denture cleaning methods was 103 denture bases with effect size at medium level (f^2^ = 0.15). The sample size for this study was sufficient.

### 2.2. Survey of Cleaning Methods

The denture cleaning methods employed were assessed by interviewing the participants and their caregivers, using a prepared questionnaire. The survey examined whether participants used a denture brush which is not a toothbrush, the frequency by which they brushed their dentures with a toothbrush or denture brush, the denture cleanser product used (if any), the frequency by which they used a denture cleanser, and the duration for which they soaked their dentures in the denture cleanser. Responses were confirmed by collating with the observation records of a full-time dental hygienist, who checked the denture brushing and denture cleansing methods used by participants in the week prior to the assessment. Denture cleanser products and denture-brush information were visually confirmed by the inspector during the oral examination. Information on the age, sex, and cognitive function of participants was provided by medical records. Cognition was only impaired in a few participants, and the extent of the decline was small.

### 2.3. Identification and Quantification of Candida spp.

To collect denture plaques, participants removed their complete dentures, which were lightly rinsed with running water to remove any saliva and then air dried. Denture plaques were collected before lunch by a single dental examiner, different from the inspector who recorded the denture cleaning information, using a sterile swab that had been immersed in phosphate-buffered saline (PBS) (Fukifuki Check II^®^, Eiken Chemical, Tochigi, Japan). The sterile swab was applied twice to the mucosal surface of the right lateral half of the examined denture. This is based on previous findings showing that the distribution of plaques on the mucosal surfaces of upper complete dentures varied between different denture regions but was bilaterally symmetrical [32], and also that *Candida* spp. were not always detected in all defined portions of the examined denture. Each sterile swab was vortexed in 10 mL PBS in a plastic bottle, the resultant samples were transported to a laboratory to be plated and incubated within 5 h of sampling, and samples were then examined to identify the *Candida* spp. present. All specimens were diluted 10-fold in saline solution and inoculated onto CHROMagar Candida plates (CHROMagarTM Candida, Kanto Chemical, Tokyo, Japan). Agar plates were aerobically incubated at 37 °C for 48 h. After the incubation period, colonies that exhibited the characteristics of *Candida* spp. (*C. albicans*, *C. glabrata*, *C. tropicalis*, and other *Candida* spp.) were presumptively identified. The colors and morphological features of these colonies were examined in more detail and the numbers of each type of colony were counted [33]. The number of colony-forming units (CFU) of *Candida* spp. per mL was assessed for each experimental denture.

### 2.4. Analysis

The total number of *Candida* spp. colonies present on experimental dentures was evaluated and expressed as log (CFU + 1) per mL. The relationship between the number of *Candida* spp. detected on the dentures and each surveyed item was assessed with Spearman’s rank correlation coefficient. Surveyed items with a probability of less than 10% in the correlation analysis were set as independent variables, and a multivariate linear regression analysis was performed. In addition, comparisons between subgroups of the surveyed items were conducted by employing the Mann–Whitney U test and Kruskal–Wallis test, together with the Dunn–Bonferroni multiple comparison test. All statistical analyses were performed by using a statistical analysis application (SPSS Statistics ver.26, IBM Japan Ltd., Tokyo, Japan). The *p*-values ≤ 0.05 were considered to be significant.

## 3. Results

### 3.1. Cleaning Methods

The number of dentures subjected to each denture cleaning method is shown in Table 1. Although only a small proportion of participants used a denture brush for cleaning (26.3%), denture cleansers were commonly used (63.8%). Regarding the frequency of use of denture cleansers, daily use was the most common (38.2%). The most commonly used denture cleansing products by participants were two alkaline, peroxide effervescent denture cleansers: enzyme-containing Polident^®^ (GlaxoSmithKline Co., Ltd., Tokyo, Japan) and Toughdent^®^ (Kobayashi Pharmaceutical Co., Ltd., Osaka, Japan). The majority of participants removed their dentures at bedtime and soaked them in a cleanser overnight.

### 3.2. Detection of Candida spp.

As shown in Table 2, *Candida* spp. were detected on 65.8% of all dentures. *C. albicans* was the most frequently detected species in the denture plaques collected and was present on 68.0% of dentures to which *Candida* spp. had adhered.

### 3.3. Relationship between the Number of Candida spp. Detected on Dentures and Cleaning Methods Employed

Table 3 shows the relationships between the number of *Candida* spp. and each surveyed item. None of the examined items related to denture brushing correlated with the number of *Candida* spp.; however, all items associated with denture-cleanser use correlated with the number of *Candida* spp. (*p* < 0.01; Table 3). The relationship between the number of *Candida* spp. and frequency of denture-cleanser use had the highest correlation coefficient (r_s_ = 0.527, *p* < 0.001). The denture cleanser product used and soaking time correlated with the number of *Candida* spp. (r_s_ = −0.306, *p* = 0.002; r_s_ = −0.291, *p* = 0.004, respectively), while sex correlated with the number of *Candida* spp. (r_s_ = −0.305, *p* < 0.001).

Comparisons between subgroups of the surveyed items that correlated with the number of *Candida* spp. are shown in Table 4. The dentures worn by female participants harbored significantly lower numbers of *Candida* spp. than those worn by male participants (*p* < 0.001). The frequency of use of denture cleansers correlated with the number of *Candida* spp. (*p* < 0.001). Dentures cleaned daily with a denture cleanser contained significantly lower numbers of *Candida* spp. than those cleaned with a cleanser once or twice a week or without a cleanser (*p* < 0.001) and had the lowest numbers of *Candida* spp. in the subgroup of the frequency of use of denture cleansers. A significant difference was observed in the number of *Candida* spp. between the different denture cleanser products (*p* < 0.01), and the number of *Candida* spp. was significantly lower on dentures cleaned with Pika^®^ (Rohto Pharmaceutical Co., Ltd., Osaka, Japan) containing a *Candida* spp.–dissolving enzyme than on those cleaned with enzyme-containing Polident^®^ (*p* < 0.01). Dentures that were soaked overnight contained significantly lower numbers of *Candida* spp. than those soaked for less than 30 min (*p* < 0.01). However, only a small number of dentures were soaked for less than 30 min (*n* = 3; see Table 1).

Table 5 shows multivariable linear regression analysis values with the number of *Candida* spp. on dentures and dependent variables, which surveyed items with a probability of less than 10% based on the correlation analysis in Table 3. In spite of this requirement for the analysis, sex as an independent variable was removed from this multivariable regression analysis because of its correlation with the frequency of use of denture cleansers and soaking times (respectively, r_s_ = 0.474, *p* < 0.001, r_s_ = −0.295, *p* = 0.001). In the multivariable analysis, the number of *Candida* spp. on dentures independently correlated with the frequency of use of denture cleansers (β = −0.553, *p* < 0.001) and soaking times (β = 0.292, *p* < 0.001), and the frequency of use of denture cleansers had the greatest impact on the number of *Candida* spp. on dentures among the items surveyed based on the standardized partial regression coefficient (β).

## 4. Discussion

The results of the correlation and multivariable linear regression analyses of denture cleaning regimens showed the strongest relationship between the frequency of use of denture cleansers and the quantity of *Candida* spp. on dentures worn by nursing home residents. These results support our hypothesis that the frequency of use of denture cleansers is related to the number of *Candida* spp. adhering to worn dentures. Furthermore, daily denture-cleanser use appeared to reduce the number of *Candida* spp. adhering to dentures; this finding is in contrast to previous findings showing that commercially available peroxide-based denture cleansers were ineffective against *Candida* spp. when used only once [23,24,25,26,27,28], and that even when used daily, they reduced the number of total microorganisms, but did not effectively remove *C. albicans* [29,30].

In an in vitro study, Ramage et al. [20] concluded that it is difficult to control established *C. albicans* biofilms via intermittent treatments with a denture cleanser. Therefore, biofilms need to be removed from dentures before they mature. This is regarded as the reason for the strong relationship between the frequency of use of denture cleansers and the number of *Candida* spp. on dentures in the present study. Therefore, among nursing home residents who used denture cleansers daily in the present study, *Candida* spp. may not have been able to mature on dentures instead of being removed well. Moreover, in an in vivo study, Ramage et al. [21] demonstrated that a daily treatment with denture cleansers was more effective at reducing total microbial numbers than intermittent treatments and also that cleansing regimens may induce compositional changes in denture plaques. A denture cleaning procedure that prevents microorganisms from colonizing dentures is considered to be more important than methods to remove them from dentures. In addition, even when the upper and lower dentures were analyzed separately, the results of multivariable linear regression analyses were the same, and the effects of the frequency of use of denture cleansers were the strongest; however, the number of *Candida* spp. detected was significantly higher on the upper dentures than on the lower dentures.

The prevalence of *Candida* spp. was previously shown to be higher among older adults who required nursing care than among independent older adults [34] and was particularly high among denture-wearing nursing home residents [7]. These findings indicate that the elimination of *Candida* spp. needs to be considered during denture cleaning by nursing home residents, in addition to the development of chemicals and mechanical cleaning methods that are effective against these microorganisms. It is difficult for nursing home residents to clean their dentures well with denture brushes; combinations of cleaning methods involving the use of chemicals and microwave, ultrasound, or light-emitting diode irradiation-based mechanical devices were recently shown to effectively remove and kill *C. albicans* [23,35,36]. These combined cleaning methods are considered to be useful for nursing home residents. However, optimal regimens have not yet been established for these combined methods. Although the daily use of these methods may be effective, the cost effectiveness and efforts required to perform them need to be considered.

Soaking times and solution temperatures in regimens for commercially available peroxide denture cleansers were previously investigated, and the findings obtained showed that soaking at room temperature for 8 h or at 65 °C for 5 min was the most effective [37]. Duyck et al. [38] reported that the overnight storage of dentures in cleaning tablet solutions effectively reduced *C. albicans* levels. These findings are consistent with the present results, which showed a correlation between denture soaking times and the number of *Candida* spp. on dentures in a multivariable analysis. However, as shown in Table 1, the majority of participants who used denture cleansers soaked their dentures at room temperature overnight (approximately 8 h). Therefore, the present study lacks uniformity in the amount of intergroup data on soaking times and was cross-sectional in nature. Nevertheless, based on the results obtained, we recommend the daily soaking of dentures overnight in a commercially available denture cleanser.

Although new denture cleaning methods may be developed in the future, further studies are needed to establish the most effective denture cleaning regimen from currently available options.

## 5. Conclusions

The frequency of use of denture cleansers correlated with the number of *Candida* spp. present on the dentures of nursing home residents. The daily use of a denture cleanser overnight is an effective method for controlling *Candida* spp. Oral care and other healthcare providers need to arrange denture cleaning environments in nursing homes accordingly and instruct and assist residents.

## Figures and Tables

**Table 1 ijerph-19-15805-t001:** Denture cleaning methods.

	Number of Dentures	
Use or not of a denture brush		
Used	40	(26.3%)
Not used	112	(73.7%)
Frequency of brushing		
3 times a day	61	(40.1%)
Twice a day	31	(20.4%)
Once a day	36	(23.7%)
Never	24	(15.8%)
Frequency of denture cleanser use		
Daily	58	(38.2%)
3–4 times a week	14	(9.2%)
1–2 times a week	25	(16.4%)
Never	55	(36.2%)
Denture cleanser product used		
Po	61	(62.9%)
To	20	(20.6%)
Pi	8	(8.2%)
Others	8	(8.2%)
Denture cleanser soaking time		
All night	94	(96.9%)
Less than 30 min	3	(3.1%)

Po, enzyme-containing Polident^®^; To, Toughdent^®^; Pi, Pika^®^ (Rohto Pharmaceutical Co., Ltd., Osaka, Japan); Others, cleansers other than Po, To, and Pi.

**Table 2 ijerph-19-15805-t002:** Detection rates of *Candida* spp. on dentures.

	Number of Detections	Detection Rate	
		Among Sampled Dentures	Among Detected *Candida* spp.
*Candida* spp.	100	65.8%	
*C. albicans*	68	44.7%	68.0%
*C. glabrata*	18	11.8%	18.0%
*C. tropicalis*	15	9.9%	15.0%
Other *Candida* spp.	26	17.1%	26.0%

**Table 3 ijerph-19-15805-t003:** Relationships between the number of *Candida* spp. on dentures and each surveyed item.

Items	Correlation Coefficient (r_s_)	*p*-Value
Age	−0.057	0.490
Sex	−0.305	<0.001 **
Use or not of a denture brush	0.149	0.068
Frequency of brushing	−0.018	0.839
Frequency of denture cleanser use	0.527	<0.001 **
Denture cleanser product used	−0.306	0.002 **
Denture cleanser soaking time	0.291	0.004 **

** Spearman’s correlation test, significant (*p* < 0.01).

**Table 4 ijerph-19-15805-t004:** Comparison of subgroups of surveyed items that correlated with the number of *Candida* spp. No significant differences were observed in the number of *Candida* spp. between subgroups marked with the same letter (Dunn’s multiple comparison test).

	Mean (SD) Log(CFU + 1/mL)	Median (Range) Log(CFU + 1/mL)	Multiple Comparison Test
Sex ^†^			
Male	2.98 (1.46)	3.31 (4.76)	
Female	1.83 (1.75)	1.81 (5.15)	
Frequency of denture cleanser use ^‡^			
Daily	0.94 (1.46)	0.00 (4.76)	a
3–4 times a week	1.83 (1.60)	2.03 (4.11)	abc
1–2 times a week	3.21 (1.25)	3.53 (4.74)	b
Never	3.02 (1.45)	3.34 (5.15)	bc
Denture cleanser product used ^¶^			
Po	2.04 (1.66)	2.28 (4.76)	A
To	1.26 (1.77)	1.41 (4.74)	AB
Pi	0.00 (0.00)	0.00 (0.00)	B
Others	1.32 (1.84)	2.06 (4.11)	AB
Denture cleanser soaking time ^§^			
All night	1.56 (1.66)	1.85 (4.74)	
Less than 30 min	4.50 (0.39)	4.69 (4.46)	

†, Mann–Whitney U test (*p* < 0.001); ‡, Kruskal–Wallis test (*p* < 0.001); ¶, Kruskal–Wallis test (*p* < 0.01); §, Mann–Whitney U test (*p* < 0.01); a, b, c, Dunn’s multiple comparison test (*p* < 0.001); A, B, Dunn’s multiple comparison test (*p* < 0.05); Po, enzyme-containing Polident^®^; To, Toughdent^®^; Pi, Pika^®^; Others, cleansers other than Po, To, and Pi.

**Table 5 ijerph-19-15805-t005:** Multivariable linear regression analysis of the number of *Candida* spp. and four surveyed items.

Independent Variables	Β	95%CI		Coefficient (*β*)	*p*-Value
Use or not of a denture brush	−0.061	−0.643	0.521	−0.017	0.835
Frequency of denture cleanser use	−1.092	−1.417	−0.766	−0.550	<0.001 **
Denture cleanser product used	−0.043	−0.167	0.082	−0.056	0.499
Denture cleanser soaking time	2.896	1.308	4.484	0.294	<0.001 **

** Significant (*p* < 0.01).

## Data Availability

The data presented in this study are available upon request from the corresponding author.

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
