# Peer review of "Effects of Denture Cleaning Regimens on the Quantity of Candida on Dentures: A Cross-Sectional Survey on Nursing Home Residents"

_ijerph, 2022, doi:10.3390/ijerph192315805_

Round 1
Reviewer 1 Report
The manuscript is well structured, comprising all the necessary parts according to the instructions for authors. It provides sound and detailed data, with a good balance of descriptive text and tables.
In the use of English, there are two minor suggestions for improvement that may be considered:
- few sentences are sometimes too long – in order to achieve better readability and understanding, clauses could be separated into several sentences – e.g. line 136-140, line 270-273;
- use of 1st person plural in lines 90-96 – although acceptable and in plural form as it refers to a hypothesis of a group of researchers, using the generally more common 3rd person form may still be considered at this point, especially as it is otherwise used throughout the whole manuscript.
As regards the content:
In the materials and methods section, the survey of cleaning methods examined also the use of a denture brush, and it was included in the results section as well. However, the discussion section focuses only on the results of the correlation and multivariable linear regression analyses, and does not mention it at all. The closing statement that „the development of new denture cleaning methods is required“ comes on rather abruptly, as the existing cleaning methods are not challenged as unsatisfactory. Therefore, this closure maybe requires some more insights.
The paper is well written and can be published with possible prior considering of minor revisions.
Author Response
Reviewer 1 ijerph-2039977
Dear Sir,
Thank you for your thoughtful responses, which have helped us to revise our manuscript. We have revised the text in response to the reviewers’ comments.
Reviewer’s comment
Reviewer 1
Comments and Suggestions for Authors
The manuscript is well structured, comprising all the necessary parts according to the instructions for authors. It provides sound and detailed data, with a good balance of descriptive text and tables.
In the use of English, there are two minor suggestions for improvement that may be considered:
few sentences are sometimes too long – in order to achieve better readability and understanding, clauses could be separated into several sentences – e.g. line 136-140, line 270-273;
Reply: Thank you very much for your review.
As you pointed out, we have split the sentences. line 143-146, line 283-287
- use of 1st person plural in lines 90-96 – although acceptable and in plural form as it refers to a hypothesis of a group of researchers, using the generally more common 3rd person form may still be considered at this point, especially as it is otherwise used throughout the whole manuscript.
Reply: Thank you very much for your review.
We have changed to the common 3rd person.
As regards the content:
In the materials and methods section, the survey of cleaning methods examined also the use of a denture brush, and it was included in the results section as well. However, the discussion section focuses only on the results of the correlation and multivariable linear regression analyses, and does not mention it at all. The closing statement that „the development of new denture cleaning methods is required“ comes on rather abruptly, as the existing cleaning methods are not challenged as unsatisfactory. Therefore, this closure maybe requires some more insights.
Reply: Thank you very much for your review.
As you pointed out, the closing statement of the discussion section was misleading and not well worded. We think that we should establish a more effective regimen in the current denture cleaning method, so we have revised the sentence. In addition, our previous study (Reference No. 23) found that denture toothbrushes are not highly effective, and because it is difficult for people requiring nursing care in institutions to use denture toothbrushes well, we have added sentences on lines 270-271.
The paper is well written and can be published with possible prior considering of minor revisions.
Reviewer 2 Report
The current study is aimed to to investigate the relationship between denture cleaning methods and/or regimens and the number of Candida spp. adhering to the dentures of nursing home residents. The topic is novel, there are a few points need to paid attention:
1) The definition of dentures in manuscript need to be clarified. The dentures means complete dentures or both complete and partial dentures.
2) The type of denture materials and fabrication techniques may contribute the difference of results. Authors need to explain them.
3) The study is human related. The design of the study needs to be approved by The Institutional Review Board, and mentioned in material and methods.
4) The examiner who collect denture plaques should be blind from questionnaire.
5) Whether the denture plaques were collected at the same time of the day needs to be mentioned. Since there maybe more plaque detected at the end of day.
6) Authors mentioned "the denture cleaning methods employed were assessed by interviewing the participants and their caregivers". So if dentures all cleaned by caregivers, how you build the relationships between the number of Candida spp. on dentures and the surveyed item age and sex.
7) It is better authors can prove the sample size is enough for current study, such as G power analysis.
Author Response
Reviewer 2 ijerph-2039977
Dear Sir,
Thank you for your thoughtful responses, which have helped us to revise our manuscript. We have revised the text in response to the reviewers’ comments.
Reviewer’s comment
Reviewer 2
The current study is aimed to investigate the relationship between denture cleaning methods and/or regimens and the number of Candida spp. adhering to the dentures of nursing home residents. The topic is novel, there are a few points need to paid attention:。
1) The definition of dentures in manuscript need to be clarified. The dentures means complete dentures or both complete and partial dentures.
Reply: Thank you very much for your review.
As you pointed out, all the places that should have been full dentures have been fixed in that way.
2) The type of denture materials and fabrication techniques may contribute the difference of results. Authors need to explain them.
Reply: Thank you very much for your review.
As you pointed out, we have corrected the dentures that used heat-cured acrylic resin instead of metal bases. In addition, it has been stated before the revision that lines 116-118 have not been relined to improve the fit of dentures. This is because the material used for relining is room-temperature polymerized resin, and it is said that the surface roughness is rough and microorganisms tend to adhere to it.
3) The study is human related. The design of the study needs to be approved by The Institutional Review Board, and mentioned in material and methods.
Reply: Thank you very much for your review.
As you noted, ethical review board approval should be mentioned in the materials and methods section. we have already written this.
4) The examiner who collect denture plaques should be blind from questionnaire.
Reply: Thank you very much for your review.
The inspector who collected the plaque was a different inspector from the inspector who recorded the questions regarding the cleaning of the dentures, so we have written it so that it could be understood.
5) Whether the denture plaques were collected at the same time of the day needs to be mentioned. Since there maybe more plaque detected at the end of day.
Reply: Thank you very much for your review.
Since the plaque collection was always between 11:00 and 12:00 before lunch, we have added that it was done before lunch.
6) Authors mentioned "the denture cleaning methods employed were assessed by interviewing the participants and their caregivers". So if dentures all cleaned by caregivers, how you build the relationships between the number of Candida spp. on dentures and the surveyed item age and sex.
Reply: Thank you very much for your review.
We selected study participants who cleaned their dentures themselves, as line 114 in the materials and methods section states that the dentures were cleaned by the users themselves. The number of dentures in the upper and lower jaws, the average age and SD of the survey subjects, and the number of genders are given in the Materials and Methods section.
7) It is better authors can prove the sample size is enough for current study, such as G power analysis.
Reply: Thank you very much for your review.
As you pointed out, we have added the sentence "we use the G power software to consider the sample size and the sample size is sufficient" in the participants section of the materials and methods section.